# Strength, Frost Resistance, and Resistance to Acid Attacks on Fiber-Reinforced Concrete for Industrial Floors and Road Pavements with Steel and Polypropylene Fibers

**DOI:** 10.3390/ma15238339

**Published:** 2022-11-23

**Authors:** Željko Kos, Sergii Kroviakov, Vitalii Kryzhanovskyi, Daria Hedulian

**Affiliations:** 1Department of Civil Engineering, University North, University Centre of Varaždin, 104, Brigade 3, 42000 Varazdin, Croatia; 2Department of Highways and Airfields, Odessa State Academy of Civil Engineering and Architecture, Didrihsona Street 4, 65029 Odessa, Ukraine

**Keywords:** industrial floors, road pavements, fiber-reinforced concrete, polypropylene fiber, steel fiber, frost resistance, corrosion resistance

## Abstract

A comparison of the effect of steel and polypropylene fibers on the strength, frost resistance, abrasion, and corrosion resistance in an acidic environment of fiber-reinforced concrete for industrial floors and road pavements was carried out. Steel fibers with a length of 50 mm and a diameter of 1 mm and polypropylene fibers with a length of 36 mm and a diameter of 0.68 mm were used. The amount of steel fiber varied from 15 to 25 kg/m^3^, and the amount of polypropylene fiber varied from 2 to 3 kg/m^3^. It has been established that steel fiber more significantly increases the concrete compressive strength, and both types of dispersed reinforcement increase the flexural strength equally by 27–34%. Also, dispersed reinforcement reduces the concrete abrasion resistance by 15–35% and increases its frost resistance by 50 cycles, which helps to improve the durability of industrial floors and road pavements. The use of steel fiber in an amount of 20 kg/m^3^ and polypropylene fiber in an amount of 2.5 kg/m^3^ also increases the concrete corrosion resistance in an acidic environment. In general, dispersed reinforcement with both fiber types has approximately the same technological effect concerning the mentioned applications. However, the use of polypropylene fibers is economically more profitable since an increase in the cost of 1 m^3^ of concrete with steel fiber reinforcement is from $22.5 to $37.5, and an increase in cost with polypropylene fiber is from $10 to $15.

## 1. Introduction and Background

Concrete industrial floors must provide a quality surface for production processes and comfortable, safe human activities. At the same time, they are operated under conditions of multidirectional loads and dynamic influences [1,2]. Rigid pavements are operated under similar conditions while additionally being affected by freezing and thawing [3,4,5].

According to the current standard in Ukraine [6], concrete grades from C12/15 to C32/40 can be used for industrial floors, depending on the load intensity. The floor material should be used in accordance with the project requirements and taking into account the reliability and durability of the structure, as well as the rational use of resources. In the regulatory documents of European countries [7] and the USA [8], similar requirements are imposed for concrete industrial floors. For rigid road pavements in accordance with Ukrainian [4] and European [5] standards, concretes of grades from C20/25 to C32/40 are used. At the same time, the main characteristic of road concrete is flexural strength, which should be from 3.0 to 4.1 MPa.

The use of fiber reinforcement is a well-known and well-established method for improving important mechanical properties for industrial floor and road pavement concretes. For dispersed reinforcement, different types of fibers are used, most often steel, polypropylene, and basalt [1,3,9,10,11]. Also, a promising option for increasing the resistance of concrete to the formation and development of micro cracks is carbon nanotube reinforcement [12,13]. However, for the quality work of carbon nanotube reinforcement, it is important to ensure their uniform dispersion in the concrete mixture. The efficiency of using each fiber type is determined not only by the value of the change in the concrete properties due to reinforcement but also by the change in the cost of concrete with dispersed reinforcement [2]. The corrosion resistance of the fiber is also important because it is impossible to protect the fibers with a sufficient layer of concrete from contact with an aggressive environment [14,15]. Alternatively, textile concretes made of carbon or AR glass fibers are increasingly used to eliminate steel reinforcement in the concrete and thus increase durability and save material [16,17].

According to many studies, reinforcement with steel fiber makes it possible to most significantly increase the concrete flexural strength and, at the same time, compressive strength. It was shown in [18] that steel fibers are superior to glass and polypropylene fibers in increasing the compressive and flexural strength even with the same volume content of dispersed reinforcement (1% of the volume of concrete of the coating). In [19], the addition of steel fibers made it possible to obtain high-strength concrete with a compressive strength of 70 MPa and improved flexural strength after post-cracking. According to [20], concrete reinforced with steel fibers has a higher flexural strength than fiber-reinforced concrete with polypropylene fibers but has lower compressive strength and workability. In [21], due to the use of 100 kg/m^3^ of steel fiber, the compressive strength of concrete pavement was increased by 10–12 MPa, and the flexural strength increased from 7–9 MPa to 15–17 MPa. However, the economic efficiency of the use of steel fiber in pavements, as well as the impact of the fiber on the environment, depends on its amount in the concrete composition [22].

The use of polypropylene fibers is also an effective method for improving the concrete properties of industrial floors and rigid pavements. For example, it was shown in [23] that dispersed reinforcement reduces the initial shrinkage of pavements and, as a result, prevents crack formation. In [24], polypropylene fiber reinforcement increased the flexural strength of concrete to 191%, while frost resistance improved by more than 25% and compressive strength increased by 28%. The positive effect of polypropylene fiber on frost resistance is also confirmed in [25]. In [26], using fiber-reinforced concrete with polypropylene fibers ensured the efficiency and safety of the pavement, including inside tunnels. According to [27], as the age of the concrete pavement increases, the positive effect of reinforcement with polypropylene fibers grows, and the maximum allowable content of fibers is 1.5% of the concrete volume. In [1,9], a positive effect of dispersed reinforcement using polypropylene fiber on the physical and mechanical characteristics of concrete industrial floors is noted. Separately, it is emphasized that the use of polypropylene fiber is more cost-effective compared to steel fiber.

In addition to the above important parameters of the concrete industrial floor’s durability, such as shrinkage, frost resistance, and abrasion, another important characteristic is the acid resistance of concrete. The authors [28] point out the importance of selecting concrete components for satisfactory operation in aggressive environments, in particular, the type of cement and microfiller. In turn, the researchers [29,30] emphasize the positive effect of steel fiber in the amount of 2% of the concrete volume on the corrosion resistance of fiber-reinforced concrete on ordinary Portland cement under the action of acid rain corrosion. Researchers [31] have established the optimal amount of polypropylene fiber (0.3% of the concrete volume) to improve the acid resistance of fiber-reinforced concrete. It should be noted that in recent years there have been very few studies on the effect of fiber material type on the resistance of concrete in an acidic environment.

Thus, comparing the effectiveness of steel and polypropylene fibers in concrete industrial floors and road pavements is a crucial task. It is important to consider the corrosion resistance of concrete reinforced with different fiber types since different substances, including acid solutions, can act on pavements. At the same time, it is important to compare the effect of different fiber types on the strength, durability, and cost of concrete [1,9,26].

## 2. Materials and Research Methods

For the fiber-reinforced concrete mixing, the following materials were used:-Portland cement CEM II/A-S 42.5 R, manufactured by CRH Ukraine in accordance with [32,33];-Granite breakstone with a maximum aggregate size of 20 mm, in accordance with [34,35,36,37];-Quartz sand with a fineness modulus of 2.75 in accordance with [35,36,37];-Polycarboxylate superplasticizer MC-PowerFlow 3200 produced by MC-Bauchemie, Bottrop, Germany, in accordance with [38];-Steel anchor fiber with 50 mm long and 1 mm in diameter, and ultimate tensile strength 1150 MPa, produced by Stalkanat-Silur, Odessa, Ukraine, in accordance [39], Figure 1a.-Polypropylene fiber "Baumesh" with 36 mm long and 0.68 mm in diameter and ultimate tensile strength 530 MPa, produced by Bautech-Ukraine (Odessa) in accordance [40] Figure 1b.

**Figure 1 materials-15-08339-f001:**
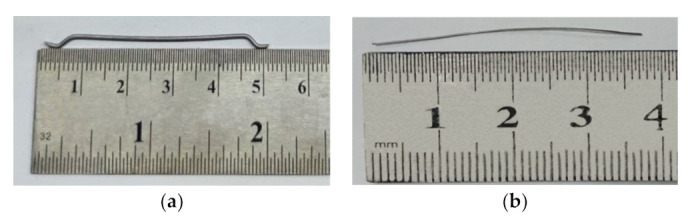
Steel anchor fiber (**a**), polypropylene fiber (**b**).

Concrete without dispersed reinforcement was studied as a control composition. Three fiber-reinforced concrete compositions with steel and polypropylene fibers were produced. The amount of steel fiber varied from 15 to 25 kg/m^3^, and the amount of polypropylene fiber varied from 2 to 3 kg/m^3^. Such a quantity of each type of dispersed reinforcement was adopted based on the recommendations of manufacturers and production practices.

The compositions of the investigated fiber-reinforced concrete with different fiber types and control concrete are shown in Table 1.

The workability of all the studied concretes and fiber-reinforced concrete mixtures was the same S4 with cone slump of 17–18 cm and determined according to [41,42,43]. To achieve equal workability, when fiber was introduced into the concrete mixture, the amount of superplasticizer was changed from 3.40 kg/m^3^ (0.94% of the cement content) to 4.08 kg/m^3^ (1.13% of the cement content). At the same time, the amount of water in the mixture remained constant for the correct comparison of the fiber-reinforced concrete properties, and the W/C ratio was 0.5.

For the preparation of fiber-reinforced concrete, a rotary mixer was used. The production and curing of the samples were carried out in accordance with the standards [44,45,46]. For each batch, 30 cubes 10 cm × 10 cm × 10 cm (6 cubes for compressive strength test at 7 and 28 days of age, 12 cubes for frost resistance test, and 12 cubes for acid attack resistance test), and 3 prisms 10 cm × 10 cm × 40 cm were manufactured. After the flexural strength test, cubes with dimensions of 7 cm × 7 cm × 7 cm were cut out of the prisms (6 specimens for each concrete mixture) for the abrasion resistance test.

For all the studied fiber-reinforced concrete, compressive strength at the age of 7 and 28 days [46,47], flexural strength at the age of 28 days [46,48,49], abrasion resistance [50,51], frost resistance [52,53] and corrosion resistance in an acidic environment [54], water absorption [55] were determined. Flexural strength was determined by a four-point load scheme (Figure 2), and load transfer was carried out at a speed of 1.77 kN/min. The determination of the frost resistance of the studied fiber-reinforced concrete was carried out according to an accelerated method [52,53]. Samples were saturated with a 5% sodium chloride solution. Control samples after extraction from the solution are tested for compressive strength. The main samples were frozen to a temperature of −50 °C for 4 h, then the temperature was raised to a temperature of −10 °C for 1 h, then the samples were subjected to complete thawing (one cycle). The main samples, after the required number of freeze-thaw cycles, are tested for compressive strength, and weight loss is also determined. The obtained results are compared with control samples, the loss of compressive strength should not exceed 5%, and the weight loss should not exceed 3%. According to the method for determining the resistance to acid attack [54], for each concrete composition, cube samples 10 cm × 10 cm × 10 cm in size were divided into two series, the first series was placed in water, the second in an acidic environment with pH = 3, obtained by adding sulfuric acid into distilled water. The samples were kept this way for 6 months, after which their properties were determined.

## 3. Research Results and Analysis

### 3.1. Compressive Strength, Flexural Strength, and Water Adsorption

It was experimentally established that the average concrete density of the base composition (no. 1) and fiber-reinforced concrete with polypropylene fiber (no. 2–4) was approximately the same and was in the range of 2415–2425 kg/m^3^. Fiber concretes with steel fibers (no. 5–7) were expected to have a slightly higher average density of 2430 to 2440 kg/m^3^.

Table 2 presents the results of testing the fiber-concrete compressive and flexural strength at 7 and 28 days.

An important indicator of the difference in the concrete structure of the control and fiber-reinforced concrete composition is that the water adsorption of concrete without fiber (no. 1) was 4.2%. For all fiber-reinforced concrete, the water absorption was lower and was in the range of 3.6–3.8%. Since water absorption is an indicator of the open porosity of concrete, it can be concluded that fiber-reinforced concretes (no. 2–7) have a lower open porosity than concrete without fibers. This can be explained by an increase in the amount of superplasticizer MC-PowerFlow 3200 when using fiber (Table 1), which affected the capillary-pore structure, as well as by the direct effect of fibers on the pores distribution and capillaries in the composite.

The effect of the steel and polypropylene fibers amount on the compressive strength is shown in Figure 3, on the flexural strength in Figure 4. The diagrams in Figure 3 and Figure 4 are built using a combined x-axis, which simultaneously shows the scale of the amount of steel fiber up to 25 kg/m^3^ and polypropylene fiber up to 3.0 kg/m^3^.

As can be seen from the diagram in Figure 3b, dispersed reinforcement increases the concrete compressive strength by 10–21% with steel fibers and by 9–17% with polypropylene fibers. Also, in the studies, it was determined that on the 7th day (Figure 3a), the concrete and fiber-reinforced concrete compressive strength was 76 to 81% of their strength at the design age. That is, dispersed reinforcement does not affect the rate of concrete hardening, which is expected.

For concrete industrial floors and pavements, the more important mechanical property is flexural strength since this property determines the bearing capacity of the pavement structure [4,5,18,27]. As shown in the diagram in Figure 4, the use of fiber reinforcement increases the concrete flexural strength by 27–34%. Improving the strength properties of concrete with fiber reinforcement can be explained by the fact that normal concrete is a brittle material with bad resistance to tensile stresses, and fiber introduction can take up internal tensile stresses by itself. The equal effect on the concrete flexural strength with less polypropylene fiber compared with a higher quantity of steel fiber interpreted that per unit of volume, there is a higher quantity of polypropylene fibers. Therefore, the distribution of tensile stresses occurs more evenly over the cross-section.

The use of polypropylene fibers increases strength by the same amount as the use of steel fibers. However, an important economic effect is that when using steel fiber its amount in fiber-reinforced concrete is from 15 to 25 kg/m^3^, and with using polypropylene from 2 to 3 kg/m^3^. The cost of steel fiber is about $1.5 per 1 kg, and the cost of polypropylene fiber is about $5 per 1 kg. Accordingly, the increase in the cost of 1 m^3^ of concrete due to reinforcement with steel fibers is from $22.5 to $37.5, and the increase in price due to reinforcement with polypropylene fibers is from $10 to $15.

### 3.2. Abrasion Resistance

Industrial floors and road pavements are subjected to abrasive action during operation. Therefore, the concrete durability of such structures is largely determined by the abrasion resistance of concrete. Figure 5 shows the effect of the amount of steel and polypropylene fibers on the concrete abrasion resistance. The lower abrasion of the material, the higher its wear resistance. Figure 6 shows samples of composition No. 5 before the abrasion resistance test.

Analysis of the diagram in Figure 5 shows that the concrete abrasion resistance of the control composition without fibers is 0.52 g/cm^2^. This level of concrete abrasion resistance does not meet the requirements of the Ukrainian national standard [56], according to which the abrasion of concrete pavement should be no more than 0.50 g/cm^2^. Due to the introduction of steel fiber, the abrasion of concrete is reduced by 15–31%, and with the introduction of polypropylene fiber by 19–35%.

Using 15–20 kg/m^3^ steel fibers or 2 kg/m^3^ polypropylene fibers makes it possible to achieve the G3 abrasion grade (0.41–0.5 g/cm^2^), which already allows the use of concrete for rigid road pavements [56]. By increasing the amount of steel fibers to 25 kg/m^3^, as well as by using 2.5–3 kg/m^3^ polypropylene fibers, the concrete abrasion grade improves to G2 (0.31–0.4 g/cm^2^), which increases the durability of road pavements and industrial floors. It is also possible to note the best wear resistance of concretes with polypropylene fibers, despite the significantly lower amount of dispersed reinforcement in concrete by weight. This can be explained by the better adhesion of polypropylene fibers to the concrete matrix compared to steel fibers. Under abrasive action, steel fibers lose their adhesion to the concrete matrix more easily, and polypropylene fibers, due to better cooperation with the matrix and retention of individual blocks of the composite, give the concrete greater wear resistance [57,58].

### 3.3. Frost Resistance

The durability of rigid road pavements in the climatic conditions of European and many other countries is significantly affected by the frost resistance of concrete [20,21,25,59]. For concrete industrial floors, frost resistance is of lesser importance, except for the floors of open warehouses. However, frost resistance is an indirect indicator of resistance to temperature fluctuations, which can occur in several technological processes.

The value of frost resistance of the studied concretes was determined in accordance with [52,53] using an accelerated method for concretes of road and airfield pavements, Figure 7. Other methods require a lot of time for the experiment and are more laborious. For example, using basic techniques according to [52,53], to achieve concrete frost resistance F100 cycles, it is necessary to carry out 100 freeze-thaw cycles; in turn, one cycle freezes at a temperature of −18 ± 2 °C and then completely thaws the samples. It is important to note that the accuracy of the accelerated technique is rather limited since it discretely distinguishes only grades F100, F150, F200, F300, etc. In general, this discreteness does not prevent us from evaluating the effect of dispersed reinforcement on frost resistance, but it does not allow distinguishing the F index in the range between F200 and F300.

It has been experimentally established that the frost resistance of control concrete composition (No. 1) is grade F150. For all other studied fiber-reinforced concretes (No. 2–9), the frost resistance grade was F200. The experimental data are presented in Table 3. Figure 8 shows specimens after frost resistance testing, (**a**): specimen with dispersed reinforcement with polypropylene fiber; (**b**): specimen with dispersed reinforcement with steel fiber.

That is, dispersed reinforcement with both steel and polypropylene fibers made it possible to increase the frost resistance of concrete to approximately the same extent. The positive effect of using fibers is explained by the ability of fibers as a stabilizing factor to prevent the separation of concrete as a composite material into separate structural blocks under the influence of freezing and thawing [20,59]. As shown above, the fiber also affects the capillary-pore structure of concrete, thereby increasing the concrete frost resistance. The influence occurs both under the action of the fibers themselves as well as changes in the dosage of the superplasticizer due to the experimental conditions. At the same time, as noted earlier, the use of polypropylene fiber due to its lower consumption per 1 m^3^ of material is more economically beneficial.

### 3.4. Resistance to Acid Attack

For industrial floors in a number of chemical industries and other enterprises using acids in the production process, as well as warehouses where substances with acidic properties are stored, an important indicator of the quality of concrete is its resistance to an acidic environment. For road pavements, acid resistance can provide concrete durability when de-icers are used.

As noted above, for all the studied fiber-reinforced concrete, the resistance to the acid attack in an acidic environment was determined. For each concrete composition, one part of the samples was kept in water for 6 months, and the second in an acidic environment with pH = 3, Figure 9. The strength of the studied concretes and fiber-reinforced concretes after 6 months of storage in an acidic environment and the magnitude of the decrease in the strength of concrete in an acidic environment are shown in Table 4. Figure 10 shows samples during testing with (a): fiber-reinforced concrete saturated with water; (b): fibrous concrete in an acidic environment, pH = 3. Figure 11 shows samples of fiber-reinforced concrete after 6 months of acid exposure with (a): steel fiber-reinforced concrete; (b): polypropylene fiber-reinforced concrete, pH = 3.

Figure 12 shows the effect of the amount of two types of fiber on the concrete strength after curing in water and in an acidic environment.

Analysis of the diagram in Figure 12a concludes that after 6 months of exposure in a humid environment, the effect of dispersed reinforcement with steel and polypropylene fibers on the concrete compressive strength is similar to the effect of fibers on strength at the standard age.

Under the influence of an acidic environment, the nature of the fiber effect on the concrete strength changes (Figure 12b). Dispersed reinforcement also has a positive effect on concrete strength and increases its value by 5–22%. However, if steel fiber showed a slightly higher efficiency in a non-aggressive water environment, then after operation in an acidic environment, the degree of different types of fiber influence on the concrete strength is equalized. It is also characteristic that in an acidic environment at low fiber dosages (15 kg/m^3^ for steel and 2 kg/m^3^ for polypropylene fibers), dispersed reinforcement has little effect on the strength and does not increase the resistance to the acid attack on concrete. The decrease in the concrete strength of the control composition was 24%, concrete with a small amount of fiber was 26% and 25%, respectively. With a higher amount of fiber, dispersion-reinforced concretes have a strength degree reduction of 23% with using steel fibers and 20–21% with using polypropylene fibers.

Such an effect of fiber on resistance to acid attack in an acidic environment is explained by the fact that as a result of the reaction of concrete components with acids, calcium carbonate is formed. It is insoluble in water and accumulates in pores and micro-cracks, which leads to concrete cracking [60,61]. In the experiment, the mass of samples 10 cm × 10 cm × 10 cm in size after exposure to an acidic environment was, on average, 15 g more than the mass of similar samples exposed to water. Dispersed reinforcement prevents the development of cracks caused by internal stresses in the composite [62]. Reducing the water absorption of concrete with fiber (open porosity) also contributes to an increase in the resistance to the acid attack of the material. But with a small amount of fiber, reinforcement is not effective enough, and internal stresses are more capable of causing destructive effects on the material. It can also be noted that the efficiency of polypropylene fibers in an acidic environment turned out to be at least not lower than steel ones, which can be explained by better adhesion to the concrete matrix and chemical inertness to acids.

In general, dispersed reinforcement with both steel and polypropylene fibers improves the concrete resistance to acid attack for industrial floors and road pavements.

## 4. Conclusions

The conducted research allowed us to carry out a comprehensive comparison of the effectiveness of concrete dispersed reinforcement for industrial floors and road pavements with steel and polypropylene fibers.

It has been established that both types of dispersed reinforcement increase the strength, frost resistance, abrasion resistance, and resistance to acid attack on fiber-reinforced concrete. Steel fibers increase the concrete compressive strength slightly more effectively. In other analyzed indicators of the concrete quality (flexural strength, abrasion resistance, frost resistance, and resistance to acid attack), the use of polypropylene fibers can improve at the same level as the use of steel fibers. To ensure increased resistance to an acid attack on concrete in an acidic environment, the amount of steel fiber should not be lower than 20 kg/m^3^, and the amount of polypropylene should not be lower than 2.5 kg/m^3^.

Dispersed reinforcement with both types of fibers increases the strength characteristics and durability of pavements under operating conditions typical of most European countries. However, due to a significant difference in the effective dosage of different fiber types, the increase in the cost of 1 m^3^ of concrete mix using dispersed reinforcement of polypropylene fibers is $12.5–$22.5 less than the increase in cost with dispersed reinforcement with steel fibers. Other advantages of polypropylene fibers compared to steel fibers are less wear and tear of technological equipment during the preparation and placing of the fiber-reinforced concrete mixture, as well as lower energy costs in manufacturing fibers.

Our data complement [29] notes that the introduction of steel and basalt fibers increased the acid resistance of concrete. Results [62] show that dispersed steel fiber reinforcement bridge cracks and inhibits the development of concrete chemical erosion. Experiment [63] reflects the positive effect of using multiscale polypropylene fiber with different geometry properties on concrete acid resistance.

It is important to emphasize that in our experiment, a more aggressive acid attack environment was used, which can be useful for expanding the range of data on the effect of dispersed reinforcement on the concrete acid attack resistance.

## Figures and Tables

**Figure 2 materials-15-08339-f002:**
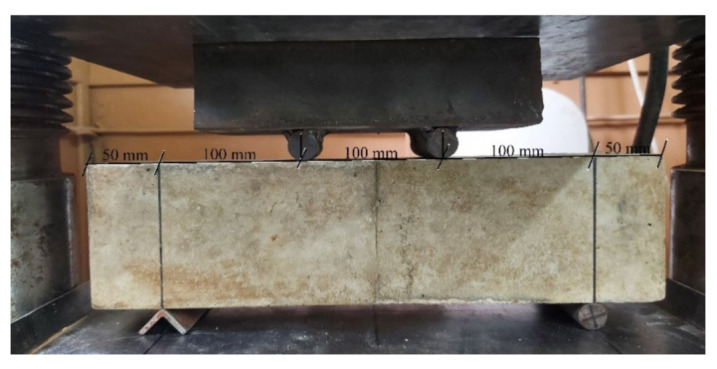
Flexural strength test using a 4-point load scheme.

**Figure 3 materials-15-08339-f003:**
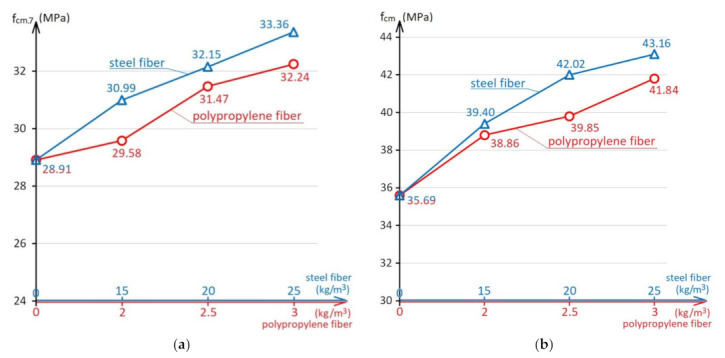
The influence of the steel and polypropylene fibers amount on the concrete compressive strength at 7 days (**a**) and compressive strength at 28 days (**b**).

**Figure 4 materials-15-08339-f004:**
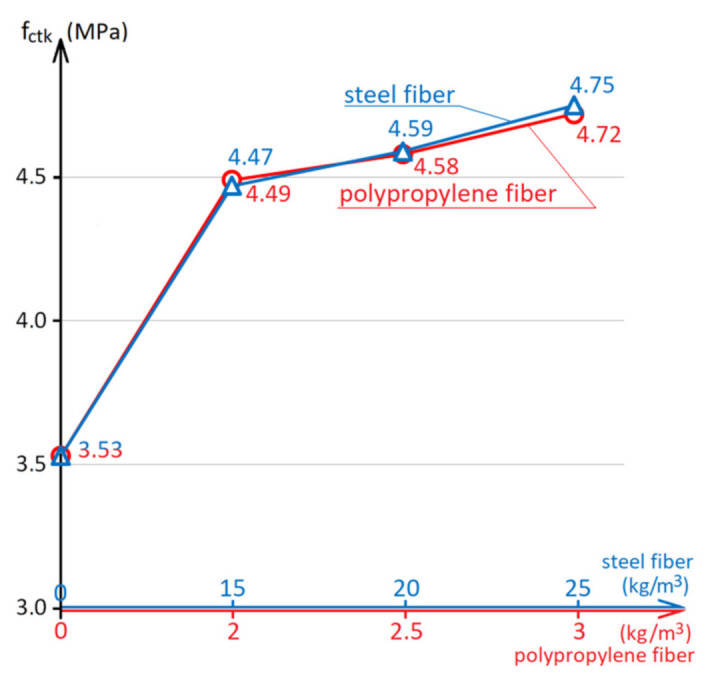
The influence of the steel and polypropylene fibers amount on the concrete flexural strength at the age of 28 days.

**Figure 5 materials-15-08339-f005:**
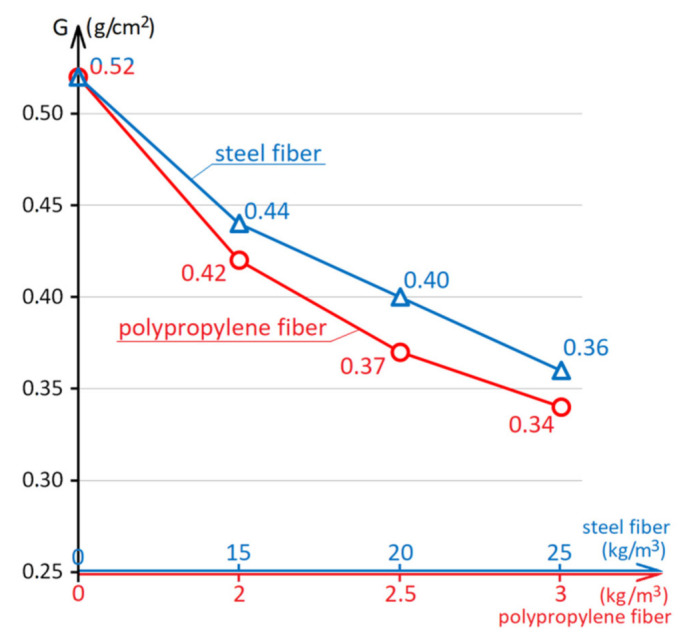
The influence of the steel and polypropylene fibers amount on the concrete abrasion resistance.

**Figure 6 materials-15-08339-f006:**
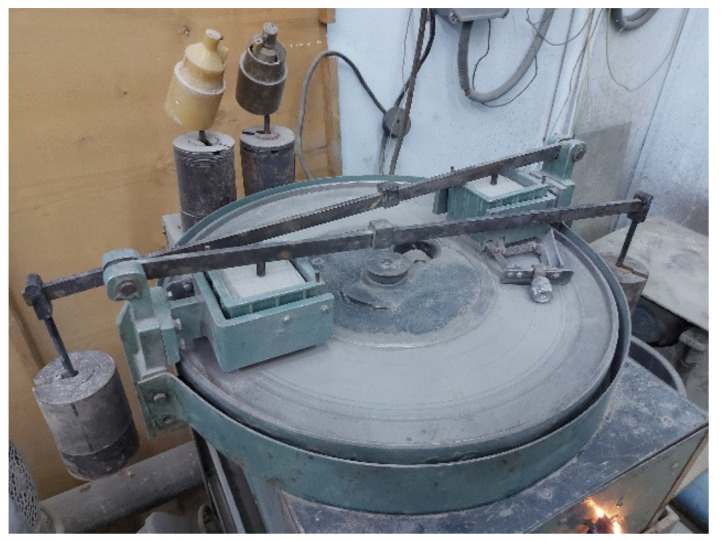
Abrasion resistance test. Composition samples No. 5 on a wheel tester.

**Figure 7 materials-15-08339-f007:**
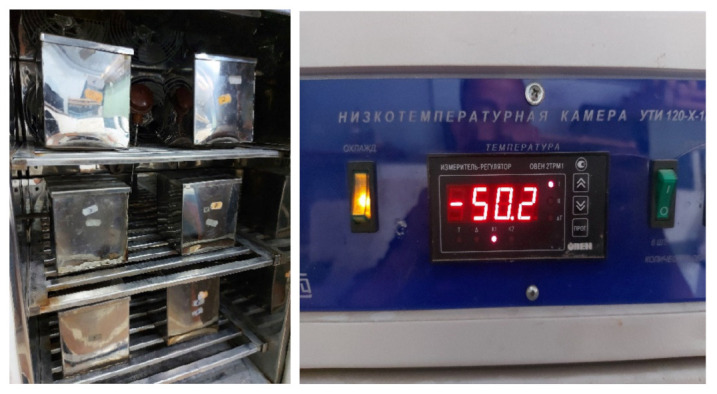
Frost resistance test.

**Figure 8 materials-15-08339-f008:**
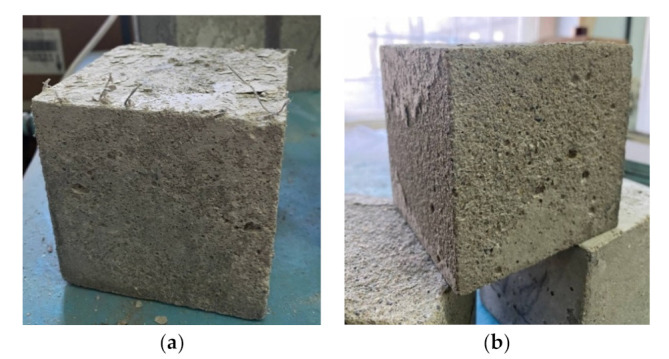
The studied specimens after freeze-thaw cycles.(**a**) fiber concrete with polypropylene fiber; (**b**) fiber concrete with steel fiber.

**Figure 9 materials-15-08339-f009:**
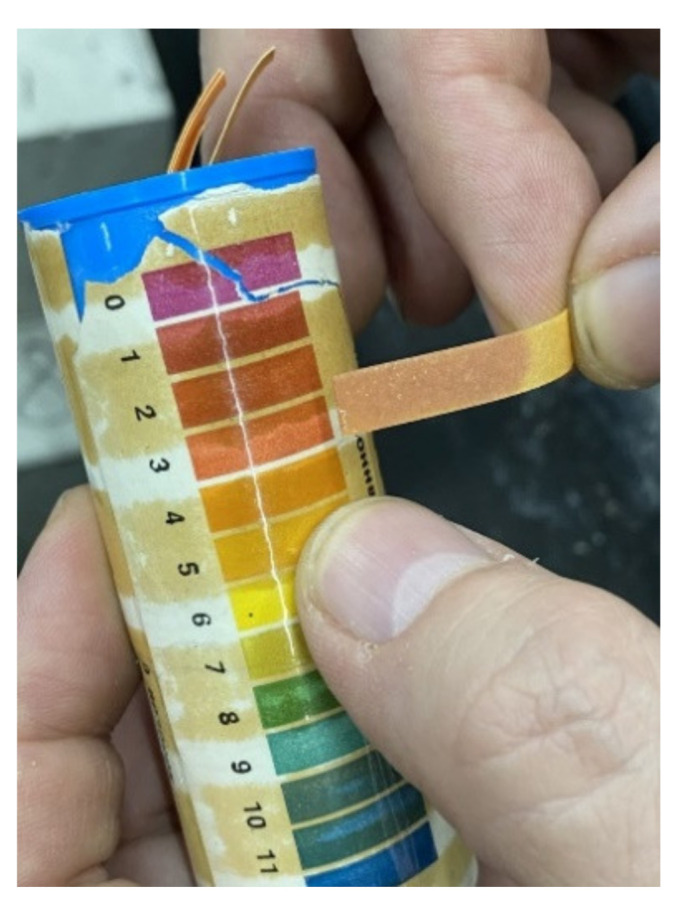
Acid environment test, pH = 3.

**Figure 10 materials-15-08339-f010:**
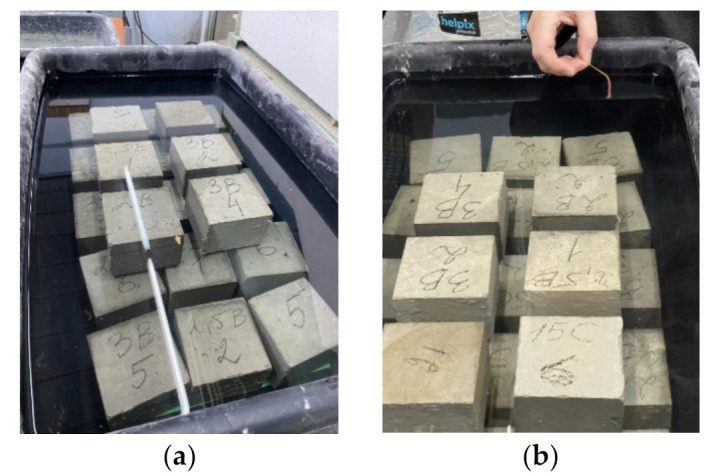
Acid environment test, pH = 3. Total amount of specimens 84 cubes (12 cubes for each batch). (**a**) control specimens; (**b**) specimens in acid environment.

**Figure 11 materials-15-08339-f011:**
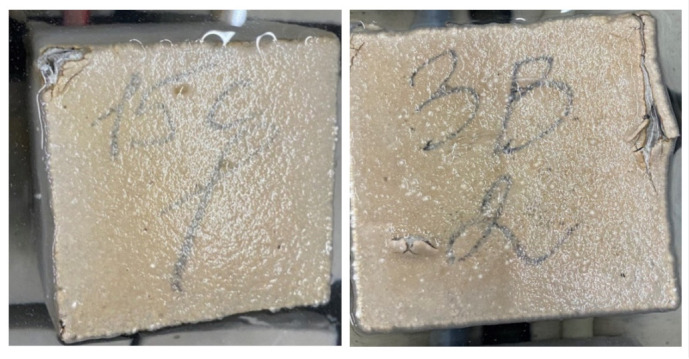
Specimens after 6 months exposure in an acidic environment, pH = 3. Total amount of specimens 84 cubes (12 cubes for each batch).

**Figure 12 materials-15-08339-f012:**
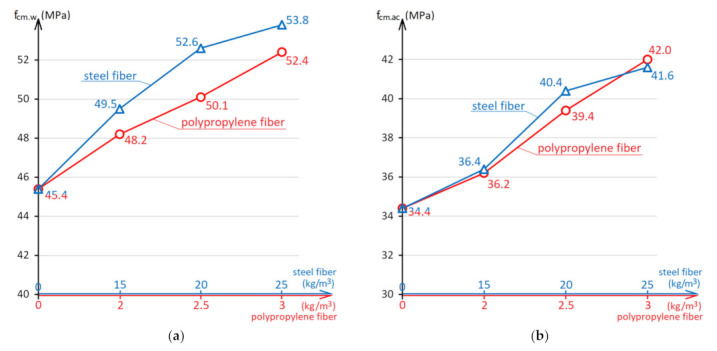
The influence of the steel and polypropylene fibers amount on the concrete strength after 6 months exposure to water (**a**) and in an acidic environment with pH = 3 (**b**).

**Table 1 materials-15-08339-t001:** Compositions of the studied concrete and fiber concrete.

No.	Marking	Compositions, kg/m^3^						
		Cement	Crushed Stone	Sand	Steel Fiber	Polypropylene Fiber	MC-Power Flow 3200	Water
1	Control concrete	360	1110	780	-	-	3.40	180
2	Fiber concrete with steel fiber 15 kg/m^3^			779	15	-	3.64	
3	Fiber concrete with steel fiber 20 kg/m^3^			778	20	-		
4	Fiber concrete with steel fiber 25 kg/m^3^			777	25	-		
5	Fiber concrete with polypropylene fiber 2.0 kg/m^3^		1105	770	-	2.0	4.08	
6	Fiber concrete with polypropylene fiber 2.5 kg/m^3^		1103	767	-	2.5		
7	Fiber concrete with polypropylene fiber 3.0 kg/m^3^		1102	763	-	3.0		

**Table 2 materials-15-08339-t002:** Values of compressive and flexural strength (‘±’ indicates standard deviation, ‘CoV’ indicates the coefficient of variation).

No. of Mixture	Compressive Strength 7 Day, MPa		-	Compressive Strength 28 Day, MPa		-	Flexural Strength 28 Day, MPa		-
	Samples	Average	CoV	Samples	Average	CoV	Samples	Average	CoV
1	28.88	28.91 ± 0.33	1.15	35.63	35.69 ± 0.79	2.0	3.54	3.53 ± 0.03	0.75
	28.60			34.93			3.50		
	29.26			36.50			3.55		
2	30.88	30.99 ± 0.86	2.73	38.96	39.40 ± 0.47	1.0	4.49	4.47 ± 0.05	1.10
	31.89			39.33			4.50		
	30.21			39.90			4.41		
3	29.55	32.15 ± 0.67	2.10	42.04	42.02 ± 0.60	1.42	4.59	4.59 ± 0.04	0.33
	28.97			42.61			4.61		
	30.21			41.42			4.62		
4	33.35	33.36 ± 0.84	2.50	43.70	43.16 ± 0.52	1.21	4.73	4.75 ± 0.04	0.53
	32.53			42.66			4.78		
	34.20			43.13			4.75		
5	29.55	29.58 ± 0.62	2.10	39.43	38.86 ± 0.63	1.61	4.48	4.49 ± 0.02	0.46
	28.97			38.19			4.51		
	30.21			38.95			4.47		
6	31.85	31.47 ± 0.52	1.65	39.91	39.85 ± 0.68	1.70	4.58	4.58 ± 0.01	0.13
	30.88			39.14			4.58		
	31.69			40.49			4.59		
7	32.34	32.24 ± 0.69	2.13	41.76	41.84 ± 0.87	2.09	4.78	4.72 ± 0.08	1.72
	32.87			42.75			4.76		
	31.51			41.01			4.63		

**Table 3 materials-15-08339-t003:** Data of fiber-reinforced concrete after frost resistance test (‘±’ indicates standard deviation, ‘CoV’ indicates coefficient of variation).

Marking of Mixture	Compressive Strength of Control Specimens, MPa			Compressive Strength of Main Specimens after 200 Cycles of Freezing and Thawing, MPa			Average Strength Reduction, %	Frost Resistance, Cycles
	Samples	Average	CoV	Samples	Average	CoV		
Control concrete	39.5	38.80 ± 1.46	3.76	36.1	36.80 ± 0.67	1.83	5.3	F150
	38.3			37.7				
	39.6			36.2				
	40.5			37.5				
	36.3			37.0				
	38.5			36.5				
Fiber concrete with steel fiber 15 kg/m^3^	40.8	40.20 ± 0.69	1.71	38.0	38.40 ± 0.46	1.20	4.7	F200
	40.5			38.2				
	40.0			39.1				
	39.1			38.9				
	39.8			38.2				
	40.9			38.1				
Fiber concrete with steel fiber 20 kg/m^3^	44.2	43.40 ± 0.84	1.94	40.9	41.40 ± 0.48	1.17	4.8	F200
	44.4			41.8				
	43.7			41.5				
	42.5			41.2				
	42.8			42.0				
	42.6			40.8				
Fiber concrete with steel fiber 25 kg/m^3^	43.9	44.70 ± 0.44	0.98	43.1	42.60 ± 0.48	1.12	4.9	F200
	44.8			42.7				
	44.8			42.4				
	44.5			42.8				
	44.8			41.8				
	45.2			43.0				
Fiber concrete with polypropylene fiber 2.0 kg/m^3^	40.5	39.30 ± 1.23	3.13	38.3	37.50 ± 0.73	1.93	4.8	F200
	40.3			37.9				
	39.5			36.2				
	38.3			37.8				
	39.6			37.4				
	37.3			37.3				
Fiber concrete with polypropylene fiber 2.5 kg/m^3^	41.6	41.40 ± 0.44	1.07	39.9	39.50 ± 0.64	1.61	4.8	F200
	41.2			39.5				
	42.0			38.8				
	40.8			38.7				
	41.7			40.2				
	41.1			40.0				
Fiber concrete with polypropylene fiber 3.0 kg/m^3^	43.3	43.30 ± 0.79	1.82	40.8	41.40 ± 0.50	1.20	4.6	F200
	44.2			41.6				
	44.1			40.7				
	43.1			41.9				
	42.9			41.5				
	42.1			41.7				

**Table 4 materials-15-08339-t004:** Strength of concrete and fiber-reinforced concrete after 6 months exposure to water and in an acidic environment (‘±’ indicates standard deviation, ‘CoV’ indicates coefficient of variation).

Marking of Mixture	Compressive Strength after Soaking in Water, MPa			Compressive Strength after Aging in an Acidic Environment, MPa			Strength Reduction, %
	Samples	Average	CoV	Samples	Average	CoV	
Control concrete	45.8	45.4 ± 0.29	0.73	34.6	34.4 ± 0.23	0.67	24
	45.6			34.4			
	45.0			34.7			
	45.0			34.2			
	45.5			34.5			
	45.2			34.1			
Fiber concrete with steel fiber 15 kg/m^3^	50.0	49.5 ± 0.35	0.72	36.3	36.4 ± 0.31	0.86	26
	49.8			36.5			
	49.1			36.6			
	49.3			36.9			
	49.5			36.1			
	49.2			36.1			
Fiber concrete with steel fiber 20 kg/m^3^	53.1	52.6 ± 0.4	0.76	40.5	40.4 ± 0.36	0.89	23
	53.0			40.9			
	52.1			40.2			
	52.5			40.1			
	52.4			40.7			
	52.3			40.0			
Fiber concrete with steel fiber 25 kg/m^3^	54.3	53.8 ± 0.32	0.59	41.3	41.6 ± 0.45	1.09	23
	54.0			41.6			
	53.6			42.3			
	53.4			42.0			
	53.7			41.1			
	53.8			41.4			
Fiber concrete with polypropylene fiber 2.0 kg/m^3^	47.5	48.2 ± 0.55	1.15	35.9	36.2 ± 0.35	0.98	25
	48.2			36.4			
	48.4			36.2			
	48.6			35.7			
	48.8			36.5			
	47.5			36.6			
Fiber concrete with polypropylene fiber 2.5 kg/m^3^	49.7	50.1 ± 0.92	1.84	38.8	39.4 ± 0.41	1.05	21
	51.1			39.6			
	49.9			39.8			
	49.1			39.6			
	49.5			38.9			
	51.4			39.5			
Fiber concrete with polypropylene fiber 3.0 kg/m^3^	52.0	52.4 ± 0.74		41.7	42.0 ± 0.55	1.30	20
	52.3		1.42	41.8			
	53.5			41.4			
	53.2			42.6			
	51.9			42.7			
	51.7			41.6			

## Data Availability

Not applicable.

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
