# Peer review of "Strength, Frost Resistance, and Resistance to Acid Attacks on Fiber-Reinforced Concrete for Industrial Floors and Road Pavements with Steel and Polypropylene Fibers"

_materials, 2022, doi:10.3390/ma15238339_

Round 1
Reviewer 1 Report
The authors conducted an experimental study to investigate the effects of steel and polypropylene fibers on the mechanical properties and durability of concrete. In general, the manuscript is well written. The following comments must be addressed to add value to the manuscript.
General Comments:
· There must be a more rigorous interpretation and scientific explanations of the mechanisms of the investigated fibers affecting the properties of concrete.
· Authors have mentioned corrosion resistance throughout the manuscript. I do not think corrosion resistance is the right term, for control and pp fiber. It is the resistance to acid attack.
· There are a lot of grammatical and punctuation errors that must be addressed. For example:
o Lines 12-13: “Used steel fiber 50 mm long and 1 mm in diameter and polypropylene fiber 36 mm long and 0.68 mm in diameter.” Please fix this sentence.
o Line 76: Please replace “growes” with “grows”
o In “Section 2. Materials and research methods” only the first letter is capital, but in “Section 3. Research Results and Analysis” each word is capitalized. Please be consistent.
o Lines 194-195: “Figure 4 shows samples of composition â„–5 before the abrasion resistance test.” Please replace “â„–5” with “No. 5”
Please summarize the main points and avoid unnecessary parts in the Abstract.
Introduction and Background is somewhat poor (incomplete) given the scope of this study. This must be improved. The most relevant knowledge attained in previous studies should be considered in this manuscript and presented, as a summary. You may discuss different types/scales of fibers like macro, micro, and nanofibers. what are their advantages and disadvantages? For example, concerning nanoreinforcement, nanofibers and nanotubes may prevent or delay the initiation of cracks at the nanoscale. The proper distribution of nanomaterials is the main issue. Microfibers, on the other hand, are easier to get evenly dispersed, but they cannot stop the initiation of nanocracks (they can only mitigate the crack propagation after the crack width reaches the microscopic scale). The following references might be considered: "Mechanical properties of carbon-nanotube-reinforced cementitious materials: database and statistical analysis", "Design and Predicting performance of carbon nanotube reinforced cementitious materials: mechanical properties and dispersion characteristics", "Probabilistic model for flexural strength of carbon nanotube reinforced cement-based materials", and "Elastic modulus formulation of cementitious materials incorporating carbon nanotubes: Probabilistic approach."
Lines 45-46: “For dispersed reinforcement, different types of fibers are used, and most often steel, polypropylene and basalt [1,3,9-11].” What do you mean with “dispersed reinforcement”? Is it “discrete reinforcement”?
Section 2. Materials and research methods: Please briefly explain the test methods. For example, with respect to flexural strength, was it three-point or four-point flexural test? Was it force-controlled or displacement-controlled? What was the loading rate? How many specimens were used for each test method? etc.
Section 3. Research Results and Analysis: I think it is better to have subsections for different properties.
Figure 2: Please mention that the compressive test results is for 7- or 28-day. Also, please have axis title for all the figures! for example; “x-axis” represents the fiber content in Figures 2, 3, and 9.
In Lines 207 and 210: Please mention the G2 and G3 abrasion grade limits.
Lines 258-262: Authors stated that “Figure 7 shows 258 samples during testing: a - fiber-reinforced concrete saturated with water; b - fibrous concrete in an acidic environment, pH=3. Figure 8 shows samples of fiber-reinforced concrete after 6 months of acid exposure a – steel fiber-reinforced concrete; b - polypropylene fiber-reinforced concrete, pH=3.” Please add these captions to Figures 7 and 8.
Table 2: Please mention Compressive strength.
Figure 2: I think it is better to have Polypropylene samples subjected to water and acid environment in one figure and the steel fiber in another figure.
As I mentioned earlier, there must be a more rigorous interpretation of the mechanisms of the investigated fibers affecting the mechanical and durability properties of the concrete. The following references might help to explain the main mechanisms in each property: "Carbon nanotube reinforced cementitious composites: A comprehensive review", "Influence of carbon nanotubes on properties of cement mortars subjected to alkali-silica reaction", and "Modeling the mechanical properties of cementitious materials containing CNTs."
Conclusions: Please organize the conclusions better. Adding bullet points with more quantitative results might help to show the findings from each topic reviewed.
Author Response
Please see the attachment.
Please see the attachment.

Reviewer 2 Report
Please make the corrections as indicated in the manuscript

Reviewer 3 Report
The authors present an interesting on improving the quality of concrete to be used in specific and harsh environment. Following are my observations.
1. Line #17, Please clarify whether the disperse reinforcement increases or decreases the abrasion resistance of concrete.
2. Line # 18, You have mentioned 50 cycles for frost resistance, however, later in the text (line #142), you have mentioned one cycle. Please add a sentence that the process is repeated for 50 times, if that is the case.
3. Line #41,42, Please explain the terminology of flexural strength, i.e., Bbtb.
4. Line # 123, what is S4, should be explained.
Round 2
Reviewer 1 Report
Authors have satisfactorily addressed the reviewer's comments.
Author Response
Thanks for the positive rating.
Reviewer 2 Report
Please make corrections as requested in review 1 and now in the second review
